



# Development of reliable future climatic projections to assess hydro-meteorological implications in the Western Lake Erie Basin

Sushant Mehan[1], Margaret W. Gitau[2*], Dennis C. Flanagan[3]

[1] Department of Agricultural and Biological Engineering, Purdue University, West Lafayette, IN 47907

[2*] Department of Agricultural and Biological Engineering, Purdue University, West Lafayette, IN 47907.

[3] USDA-Agricultural Research Service, National Soil Erosion Research Laboratory, 1196 Building SOIL, Purdue University, 275 S. Russell Street, West Lafayette, IN 47907-2077 USA

*Correspondence to*: Margaret W. Gitau, Ph.D. (mgitau@purdue.edu)

**Abstract.** Modeling efforts to simulate hydrologic processes under different climate conditions rely on accurate input data; inaccuracies in climate projections can lead to incorrect decisions. This study aimed to develop a reliable climate (precipitation and temperature) database for the Western Lake Erie Basin (WLEB) for the 21$^{st}$ century. Two statistically downscaled bias-corrected sources of climate projections (GDO and MACA) were tested for their effectiveness in simulating historic climate (1966-2005) using ground-based station data from the National Climatic Data Center (NCDC). MACA was found to have less bias than GDO and was better in simulating certain climate indices, thus, its climate projections were subsequently tested with different bias correction methods including the power transformation method, variance scaling of temperature, and Stochastic Weather Generators. The power transformation method outperformed the other methods and was used in bias corrections for 2006 to 2099. From the analysis, maximum one-day precipitation could vary between 120 and 650 mm across the basin, while the number of days with no precipitation could reduce by 5-15 % under the RCP 4.5 and RCP 8.5. The number of wet sequences could increase up to 9 times and the conditional probability of having a wet day followed by wet day could decrease by 25%. The maximum and minimum daily air temperatures could increase by 2-12 % while the annual number of days for optimal corn growth could decrease by 0-10 days. The resulting climate database will be made accessible through an open-access platform.

## 1 Introduction

Predictive hydrologic studies require accurate weather input to simulate hydrologic processes within a watershed (Obled et al., 1994). Any inaccuracies or bias associated with the weather data may lead to deleterious effects on simulated outputs (Kouwen et al., 2005;Obled et al., 1994;Shrestha et al., 2004). As a rule of thumb, the better the input climate data, the more reliable the outcomes of modeling studies can be. Such modeling outcomes can help the stakeholders or decision makers to formulate pollution mitigation strategies. Transport of pollutants as well as their dilution by water flows are also dependent upon climate (Whitehead et al., 2006). Moreover, studies based on impacts on hydrological processes due to changing climate have become possible using results from simulations from large scale general climate models. However, climate projections at regional scales are unclear and suffer from some bias because of the influence of local factors (Wilby and Wigley, 1997;Wilby and Wigley, 2002;Wilby et al., 2004). These local factors include topography and catchment



characteristics, atmospheric circulation, and moisture supply (Bosshard et al., 2014;Wild et al., 2008), and usually produce errors or bias within climate values, which may alter the outputs of many different model application studies.

For the Great Lakes Region, and in particular for the WLEB, data for several Representative Concentration Pathways (RCPs) (RCP 2.6, RCP 4.5, RCP 6, RCP 8.5) scenarios from different GCM (General Circulation Model) models at varied

resolutions (100km-600km) are available (Winkler et al., 2012;Maurer et al., 2014;Wang and Kotamarthi, 2015). These cannot be directly used for hydrologic studies due to their coarse resolution and large uncertainties associated during downscaling. These coarser resolution products from GCMs need to be resolved into finer resolution Regional Climate Models (RCMs), which is achieved using different downscaling techniques (statistical and dynamic), which are discussed in greater detail in subsequent paragraphs.

In statistical downscaling, the relationship between large scale climate variables from GCMs (predictors) is determined using fine scale climate variables for RCM (Wilby et al., 2004). Statistical downscaling is computationally inexpensive, requires less time, and involves different methods to produce the projections (Wilby et al., 2004). On the other hand, dynamic downscaling techniques develop an RCM that is derived from a GCM with the same set of empirical equations and physical principles that were used to develop the GCM (Wilby and Wigley, 1997;Xu, 1999). The outputs are resolved at a resolution

less than 50 km and can be used for regional studies at the catchment scale (Teutschbein and Seibert, 2010;Teutschbein et al., 2011). There is a major limitation with simulated outputs from dynamic downscaling; a dynamically simulated RCM may not be applicable to locations other than the region for which it was developed (Trzaska and Schnarr, 2014). In either downscaling approach, there may be the need for post-processing of the projected output from a downscaled GCM (Eden et al., 2014) to correct for bias in the data. Some errors associated with baseline climate data (Beven, 2011) and many of the

natural variabilities and uncertainties including future greenhouse gas emissions, the structure of climate models and their parameterization, and downscaling techniques (Kay et al., 2009) are not easy to simulate with sufficient or viable model runs based on computations and resource availability, which may produce some biases (Teutschbein et al., 2011). Bias-correction and perturbation are usually performed to correct or remove the bias or biases.

Bias-correction and perturbation are some post-processing options after downscaling (Troin et al., 2015). Bias-correction

helps to maintain the statistical relationships between the distributions of observations and model outputs of different climate variables for the current period simulated along with future period (Troin et al., 2015). The perturbation approach assumes that change in the distribution of observations from current to future will be the same as the model distribution (Ho et al., 2012).

Different bias-correction techniques can lead to different results in climate change impact studies (Seguĩ- et al.,

2010;Teutschbein et al., 2011). Therefore, it is very important to quantify the bias in outputs generated from the climate models before they are applied in climate change impact modeling studies (Teutschbein et al., 2011). Different sources of



uncertainty arising from GCM or RCM structure and hydrological model parameterization have been studied but evaluation of GCM and RCM model outputs of different climate variables for climate change impact studies are rarely studied (Dobler et al., 2012), specific to the WLEB. Previous climate change implication studies in the WLEB have used projected daily climate data summaries from different sources without quantifying the bias associated with it (Cousino et al., 2015;Kalcic et al., 2016;Scavia et al., 2016). This study addresses the gap where reliable climate information for simulating future water resource responses in the WLEB is lacking. The goal was to develop a framework to evaluate and correct biases associated with simulated weather output from the most reliable and easy to access statistical downscaled models available in the public domain for the WLEB, and produce a reliable climate database for the entire WLEB for 2006-2099.

Three major research questions were answered in this study. First, we assessed two sources of climate projections available in the public domain and selected the one having less bias for further analysis.  Second, we evaluated the performance of different methods in correcting the bias of the climate values obtained from the source selected for the historical period. Finally, future climate values for the 21st century were developed using the most effective bias-correction method for the eight stations in the WLEB.

## 2 Materials and Methods

### 2.1 Study Site and Climatology

The WLEB extends into Michigan, Indiana, and Ohio, and is a 29,137 km$^2$ spatially wide watershed that drains into Lake Erie, the shallowest of the five Great Lakes (Fig 1). Total annual precipitation varies from 1050 to 1200 mm (1966-2015), with more occurring during the spring season. To answer the research questions in this study to select the climate projection source and evaluate bias-correction methods, three of the eight stations were used to develop methodology (Adrian, MI, Fort Wayne, IN, and Norwalk, OH). The three stations were selected based on their geographical location and difference in precipitation, and best represent the tristate area. Moreover, the magnitude of precipitation events was different for the three stations both spatially and temporally. Norwalk received relatively greater precipitation event depths (0-40 mm) with less frequency than the other two stations (transparent to green as seen in the color pallet/legend in Figure 2 besides the three stations). However, Adrian and Fort Wayne received more frequent precipitation events with lesser depths (0-20 mm). The temporal spread of precipitation was quite variable and with the diverse geographic coverage, these three stations were considered satisfactory to answer the first two major questions of this study.





Figure 1. Pilot site: The Western Lake Erie Basin (WLEB) and different ground-based climate stations considered in this study





Figure 2. Temporal distribution of daily precipitation records (mm), for Adrian, MI; Fort Wayne, IN, and Norwalk, OH from 1966 to 2015 from the respective ground-based weather stations. Red boxes encompass the greatest magnitude precipitation event for each station.



## 2.2 Data Acquisition

Downscaled climate data for this project were obtained from two sources: 1.) GDO (authors created acronym for Global Downscaled Climate and Hydrology Projections), available at the URL: https://gdo-dcp.ucllnl.org/downscaled_cmip_projections/; and 2.) MACA (Multivariate Adaptive Constructed Analogs), available at the

URL: https://climate.northwestknowledge.net/MACA/. The historical period for both sources was 1950-2005 and future climate projections ranged from 2006-2099. Data from both sources were statistically downscaled from the set of GCMs and have been used in many climate change studies (Cousino et al., 2015;Ficklin et al., 2009;Mehan et al., 2016). The sources provide fine spatial resolution translations of climate projections over the United States based on the multi-model dataset referenced in the IPCC AR 5 (CMIP5) to an extent of 0.25 and 0.04 degrees, sufficient for regional climate impact

assessment studies.

The GDO source incorporates non-dynamic approaches including monthly Bias-correction and Spatial Disaggregation (BCSD) and daily Bias-corrected and Constructed Analogue (BCCA), which have been well tested and automated to produce output statistics matching those of a historical period for fine scaled gridded precipitation and temperature (Abatzoglou, 2013). Under the BCSD method, quantiles of historical patterns are related to quantiles of predictions from the

GCM to project daily time series for the downscaled grid. GCM predictions are matched statistically with a set of observed historical weather patterns to develop the fine scale map while downscaling using the BCCA method. The drawback of using GDO downscaled data is the assumption that the statistical properties of the high resolution GCM and local scaled RCM after downscaling, including mean and variance are constant through time, which is not the true case (Brekke et al., 2013;Wood et al., 2004).

The other source was the Multivariate Adaptive Constructed Analogs (MACA) dataset, in which both the observation dataset and GCM outputs are resolved to either 4 km or 6 km. To overcome the problem of limited availability of suitable weather analogues in changing climate, seasonal and yearly trends at each grid point are computed using 21 days, 31-year running mean of data. A cumulative distribution function (CDF) of 15-days is computed at each grid point using non-parametric quantile mapping, and the CDF of historical data is used for bias-correction. The final outputs are consistent with the GCM

data and compatibility with the observational dataset is ensured. Downscaled variables include 2-m maximum and minimum temperature, 2-m maximum and minimum relative humidity, 10-m zonal wind, downward short-wave radiation, 2-m specific humidity, and precipitation accumulation all at a daily time step. There are two versions of MACA data and the difference between them pertains to epoch adjustments for variables and periods, while removing the trend at the start. For this study, MACA version 2 was used.

The two sources provide outputs from different GCMs under different RCP scenarios. GDO simulate values for RCPs 2.6, 4.5, 6, and 8.5 from 40 GCMs, whereas MACA has output from 20 GCMs, for RCP 4.5 and 8.5. For this study, nine GCMs were selected for preliminary assessment (Table 1), and all were available from both GDO and MACA. Analysis, comparisons and evaluations were performed using climate projections from 1966-2005 for GDO and MACA and observed



ground-based weather station data. Prior analysis of the Ft. Wayne station (Mehan et al., 2017a) indicated an increasing trend in precipitation depth from 1966 forward, thus we selected 1966 as the beginning year in our analyses.

**2.3 Data Analysis**

The dataset for the historical period obtained from the two sources was compared to the observed data from the three ground-based climate stations for 1966-2005. Data analysis included comparisons of means and distributions of the observed data and simulated values. Beyond the descriptive statistics, we computed precipitation conditional probabilities, various climate indices, model performance coefficients, and verification forecasting and skills scores to evaluate the performance of each data source and different methods of bias-correction. Comparisons were performed to quantify the error in simulated values in terms of their distributions, descriptive statistics, and extremes, including climate indices. A list of all the climate indices, verification forecasting and skill scores is provided in Table 2 with definitions and applications. For this study, a day with precipitation depth less than 0.1 mm was considered a dry day and any day with precipitation depth $\geq 0.1$ mm was considered a wet day.

The analysis began by comparing GDO and MACA values with the observed data for the three ground-based stations from the National Climatic Data Center (NCDC). The one source that performed better in simulating climate values was selected and then treated with different bias-correction methods, that were chosen after extensive review of literature (Leander and Buishand, 2007;Leander et al., 2008;Teutschbein et al., 2011), with care taken to preserve the means and variances. One method was a conventional one that included power transformation (Leander and Buishand, 2007;Leander et al., 2008) and variance scaling of temperature (Chen et al., 2011a;Chen et al., 2011b). The other bias-correction method was novel and based on conclusions and discussions from previous studies (Guo et al., 2017;Mehan et al., 2017a), where Stochastic Weather Generators (SWGs) performed better at simulating greater depths of precipitation. We postulated that SWGs could be used to redistribute the precipitation and simulate greater daily precipitation depths, which otherwise would be distributed to dry days or days with lower or no precipitation, adversely affecting the simulation outputs from crop growth and hydrologic models.

To evaluate the performance of SWGs for bias-correction, the climate values from the better performing climate projection source were used as an input to two SWGs: CLimate GENerator (CLIGEN) (Nicks et al., 1995) and the Long Ashton Research Station Weather Generator (LARS-WG) (Semenov, 2010;Semenov and Barrow, 2002). The weather generators were used in their default state without changing their parametrization for the historic period for analysis. Twenty-five different realizations (Guo et al., 2017) were generated for all nine GCMs at the three stations, to capture the variability and correct for bias or reduce error. Since the interest was to redistribute the precipitation to capture the high magnitude precipitation events, the extreme percentiles (75th and 90th) from the 25 different realizations were used for precipitation depth comparisons and means were used for temperature comparisons. This was because precipitation is not normally distributed but temperatures are. The 75th percentile or interquartile range (as 0th percentile was zero) and 90th percentile



were thought to pick up the extreme precipitation events well which were not captured by GCMs. Moreover, maximum variation was noticed at higher percentiles while simulating greater precipitation depths using SWGs (Mehan et al., 2017a).

After the evaluation of the different bias-correction methods, the best approach was used to develop the correction factors using the historic period and was translated to the climate projections from the different climate models for the eight stations in Figure 1. The reliable climate projections, so generated, can be used for understanding changing climate impacts on water resources in the WLEB. Methodology in this study can be extended to any study site.

## 3 Results

### 3.1 Comparison of two different climate projections (GDO and MACA)

The density distribution plots for count of events having magnitude equal to monthly precipitation total events in each year for the period from 1966 to 2005 (Fig 3a) showed that performance of both climate projection sources was similar. The GCMs distributed the corresponding amounts of precipitation to the dry days or days simulated with high precipitation depths. Therefore, there were some model outputs which simulated more counts of events having monthly precipitation totals between 20-100 mm, than what was seen for the observed data (highlighted in red box in Fig 3a). Some models were even over estimating the events having monthly precipitation totals more than 100 mm in either case. MACA had a wider range of simulation outputs from the different GCMs compared to GDO. The MACA GCM output did not perform well during late 1970s and 80s in capturing lower values of precipitation totals seen in the observed data (red boxes in Fig 3b), whereas around mid-1970s and late 1990s, the GDO projections did not capture higher values of precipitation total events (Fig 3b) for Adrian, MI.

Descriptive statistics from all different GCMs from the two different climate projection sources showed that both the sources performed equally well in simulating mean, skewness, and kurtosis for all three stations for daily precipitation depth, maximum air temperature, and minimum air temperature (Supplementary Table S1, S2, S3). However, GDO did not perform as well in simulating the number of dry days when compared with the MACA outputs (Table 3). The nine GCMs from GDO simulated one-day maximum precipitation values from 65.4 mm to 110.1 mm when compared to 120.4 mm from the observed data for Adrian, MI. The daily air temperature analysis revealed that both the sources performed at par in simulating the descriptive statistics for maximum and minimum air temperature for all three stations (Supplementary Table S2 and S3) with the exception that MACA values overestimated the number of days with maximum temperature greater than 35°C (Table 3). Descriptive statistics of one-day maximum precipitation for Fort Wayne, IN suggested that the mean value of one-day maximum precipitation was not well simulated by GDO climate models. The results for performance evaluation in simulating one day maximum precipitation by two different climate projection source (GDO and MACA) can be seen in Supplementary Table S4. There were less dry days simulated from climate models from GDO for Fort Wayne and Norwalk



for almost all months in a year when compared with the observed values from respective ground-based climate stations (Supplementary Table S6).



Figure 3. (a) Density distribution charts for Adrian, MI for count of monthly precipitation depths, mm; (b) Distribution of annual precipitation depths, mm, with range bounds from different GCM outputs (For Fort Wayne, IN and Norwalk, OH, please refer Supplementary Figures S1 (A) and (B)).

On the other hand, the number of wet days in a month as simulated by the GDO source were relatively much greater than what was simulated by the MACA source for all three stations in a month (Supplementary Table S5). The extreme event analysis or climate indices showed that maximum dry length was overestimated by GDO for all three stations (Supplementary Table S7). This strengthens the previous observation that GDO climate models were over estimating the number of precipitation days (Supplementary Table S1), e.g., in the case of Adrian, MI (Fig 4). On the other hand, the





maximum wet length was over simulated by both climate projection sources but greater discrepancies in terms of wider ranges and average ensemble values were seen with GDO outputs. The numbers of dry sequences were under estimated and wet sequences were over simulated by both GDO and MACA for all three stations, as presented for Adrian, MI in Figure 4. The average output values from nine different climate models for maximum dry length for Norwalk, OH were 12 and 21

5 days from GDO and MACA, respectively, corresponding to 25 days recorded from the observed data (Supplementary Table S7). On the other hand, 44 and 20 days were simulated as the maximum wet length from GDO and MACA simulation outputs, respectively, for Norwalk, OH, compared to 18 days from observed data (Supplementary Table S7). For Fort Wayne, IN, the GDO source output estimated 72 days for optimum corn growth, while MACA source output estimated 65 days; observed was 63 days. On the other hand, for Adrian, the average value from the outputs of nine different climate

10 models while simulating snow days with GDO and MACA sources were both 50, compared with the observed 30 days (Fig 4).

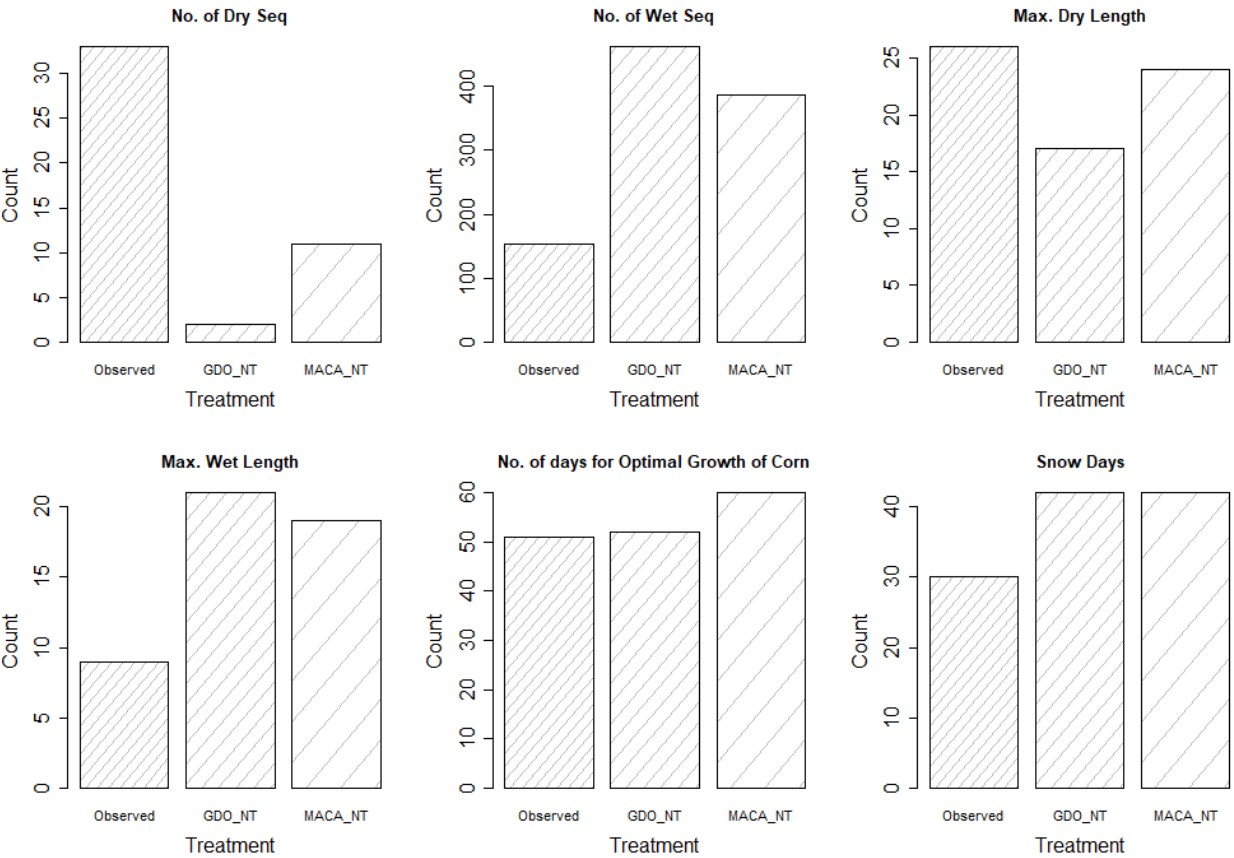

Figure 4. Comparison of GDO and MACA climate projection sources while simulating different climate indices for Adrian, MI between 1966 and 2005 (GDO_NT: GDO No Treatment; MACA_NT: MACA No Treatment). Plots for Fort Wayne, IN

15 and Norwalk, OH can be seen at Supplementary Figure S3 (A) and (B).



The transitional probabilities values from GDO outputs for all three stations indicated that the climate dataset lacked much needed accuracy (Table 4). The mean lengths of the dry and wet periods were underestimated and overestimated, respectively, for all three stations and the pattern reversed for return period in years to have an event equivalent to the mean lengths of the dry and wet periods from GDO and MACA (Table 4). Such outcomes strengthen the argument that because

the mean length of the dry period simulated using GDO was almost double than what was recorded in the observed data, the return period to have a dry event was so long for the GDO projections, whereas the MACA projections were more reasonable (Table 4). Verification forecasting or skill scores along with performance coefficients for precipitation events showed that GDO had a higher Brier score (0.6) than MACA (0.5) for Norwalk, OH; which indicated that GDO projections were relatively more offset from the observed data unlike MACA (Fig 5). GDO source had less percent correct (0.4) in

comparison to MACA (0.5) for Norwalk, OH. Not much information was observed in evaluating the performance of the two different projection sources using Gini coefficient, LEPS Score, HSS, and PSS as all the values showed that both GDO and MACA lack skill in projecting the climate data (Supplementary Table S8). Higher bias was recorded for GDO (1.7, 1.8, and 2.1 for Adrian, Fort Wayne, and Norwalk), when compared with 1.4, 1.2, and 1.3 for the MACA climate source for the three stations, respectively. Lower EDS score for MACA showed that there was greater dependence between the projected GDO

output and observed data and the GDO forecast was less random. The GDO EDS scores for Adrian, Fort Wayne, and Norwalk were 0.3, 0.4, and 0.6, and for MACA were 0.2, 0.1, and 0.2, respectively. No high correlations were seen between projected values from either source and the observed data. Greater NSE for MACA outputs indicated better performance of the MACA over GDO (Supplementary Table S8). Relative performance of both climate projection sources stayed the same while simulating the growth degree days for all stations (Supplementary Table S9). From the outputs discussed above, out

of the two given sources of climate projection for the historic period, the MACA source performed better than GDO in most of the parameters. Despite similar conditional probabilities for both sources, GDO overestimated the median daily precipitation and underestimated the days with no precipitation, resulting in more wet days. Therefore, the MACA source was evaluated for further analysis to correct its biases in the subsequent section using different methods.

### 3.2 Evaluation of different bias-correction methods for the historic period

The climate values from different climate models from the MACA source were treated with different methods of bias-correction, including conventional methods (power scaling for precipitation and variance scaling of temperature) and SWGs. The Q-Q plots between simulated values and observed data for Fort Wayne revealed that the conventional method redistributed the precipitation and simulate higher values for maximum daily precipitation (Fig 6). The LARS-WG 90th percentile approach produced better maximum daily precipitation depths for Fort Wayne and daily precipitation depths for

all three stations than the LARS-WG 75th, CLIGEN 90th, and CLIGEN 75th percentile approaches, but not as good as the power transformation of precipitation (conventional method) (Fig 6). The output values from the SWGs simulations did not perform well in simulating values daily precipitation depth. The performance of the LARS-WG and CLIGEN outputs were better in one or the other case. It can be argued for use of higher percentiles than the 90th percentile from the weather



generators, as these might result in better results or using the entire set of 25 realizations than just a single time series from 25 different realizations, which would be evaluated in future study. Precipitation suffered from the most bias (evident in section 3.1), while maximum and minimum temperature did not require much bias-correction (Fig 7 and 8). The Q-Q plots drawn from outputs of different climate models when treated with different bias-correction methods, showed that the SWGs

and Variance scaling of temperature did not perform well in correcting biases; rather the default projected values from either GDO or MACA suffered from less bias (Fig 7 for Adrian, MI) and can be used without any bias-correction for studies based on temperature projections.

Furthermore, descriptive statistics computed at a daily time step (Supplementary Table S1, S2, and S3) and seasonal basis (Table 5) were also evaluated to compare the performance of different bias-correction methods. For all three stations, mean,

skewness, and standard deviation were captured well with a conventional method of bias-correction (power transformation). The LARS-WG 75th percentile was able to project mean values better, however, standard deviations were simulated better with LARS-WG 90th percentile output (Supplementary Table S1). Average values for daily precipitation from nine different climate models for mean, skewness, and standard deviation were: Adrian: 2.4 mm, 5.5, 6.6 mm; Fort Wayne 2.5 mm, 6.7, 5.3 mm; and Norwalk 2.6 mm, 7, 7.3 mm after conventional bias-correction method, while the values from the observed data

were 2.4 mm, 4.9, and 6.5 mm; 2.5 mm, 4.7, and 6.7 mm; and 2.6 mm, 7.5, and 7.0 mm, respectively.  The maximum values of daily precipitation were also improved after using the power transformation. The average value of maximum daily precipitation from the different climate models was 220.7 mm for Norwalk, OH; 108.8 for Fort Wayne, IN; and 125.1 mm for Adrian, MI when compared to observed data of 229.1, 111.8, and 120.4 mm, respectively (Supplementary Table S1).

Seasonally, the predicted number of days with no precipitation by all the treatments were underestimated when compared

with ground-based observed data. The maximum one-day precipitation was overestimated after power transformation, only for Adrian, MI. It was 125.1 mm for Adrian, MI when compared with ground-based station value of 120.4 mm. For all other treatments and across all other stations, the maximum one-day precipitation simulated was underestimated, though the best results was achieved with power transformation method over SWGs (Supplementary Table S10, S11, and S12). The average value for maximum precipitation simulated by all climate models for Adrian after conventional bias-corrections was 102.5

mm when compared with 80.3 mm recorded from the observed data during spring; for Norwalk, OH, the average value of one day maximum precipitation during spring after power transformation was 146.7 mm compared to 87.6 mm from the observed data (Supplementary Table S12).The average value of maximum daily precipitation from the climate models after the conventional bias-correction method was 108.5 mm compared to 71.9 mm from the ground-based station data for Fort Wayne, IN during summers (Supplementary Table S11).





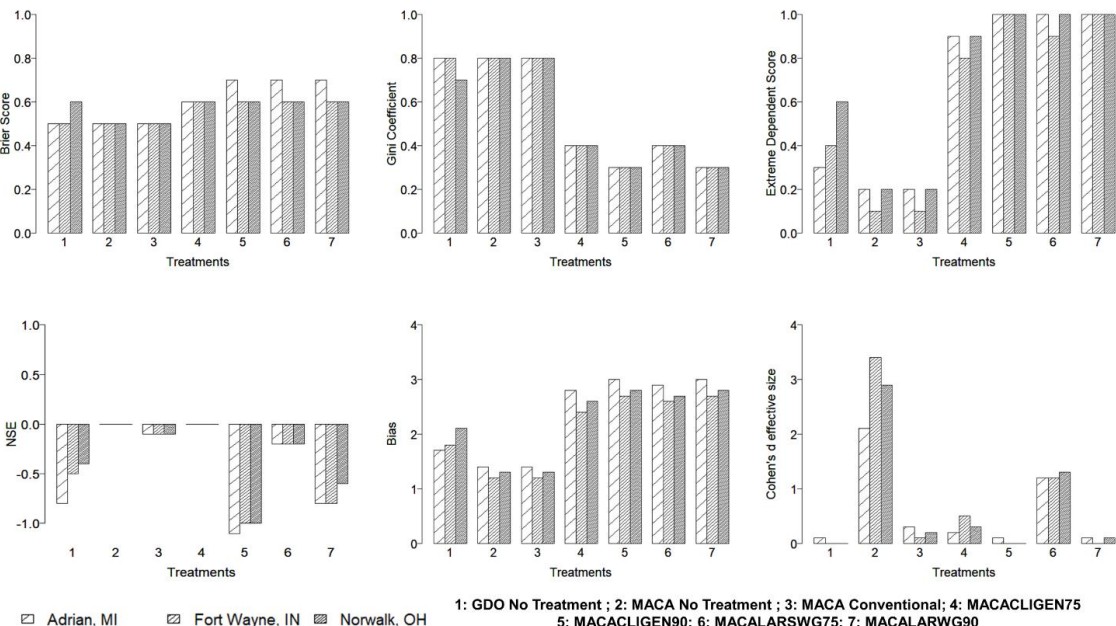

Figure 5. Comparison of performance GDO and MACA climate projection sources without any bias-correction and MACA climate source with different bias-correction treatments using verification forecasting or skill scores for period between 1966 and 2005.







Figure 6. Q-Q Plots to evaluate the performance of different bias-correction methods for period between 1966 and 2005 to
5   reduce the bias in simulating values for daily precipitation, mm and to present the future climatic scenarios (2006-2099) for
Fort Wayne, IN. (Note: Q-Q plots to evaluate performance of different bias-correction methods for period between 1966 and
2005 to reduce the bias in simulating values for daily precipitation, mm and to present the future climatic scenarios (2006-
2099) for other two stations can be seen in Supplementary Figures S2 (A) and (B)).





Figure 7. Q-Q Plots to evaluate the performance of different bias-correction methods for period between 1966 and 2005 to reduce the bias in simulating values for daily maximum temperature, °C and to present the future climatic scenarios (2006-2099) for Adrian, MI. (Note: Q-Q plots to evaluate performance of different bias-correction methods for period between 1966 and 2005 to reduce the bias in simulating values for daily maximum temperature, °C and to present the future climatic scenarios (2006-2099) for other two stations can be seen in Supplementary Figures S2 (C) and (D)).





Figure 8. Q-Q Plots to evaluate the performance of different bias-correction methods for period between 1966 and 2005 to reduce the bias in simulating values for daily minimum temperature, °C and to present the future climatic scenarios (2006-2099) for Norwalk, OH. (Note: Q-Q plots to evaluate performance of different bias-correction methods for period between 1966 and 2005 to reduce the bias in simulating values for daily minimum temperature, °C and to present the future climatic scenarios (2006-2099) for other two stations can be seen in Supplementary Figures S2 (E) and (F)).





The climate indices analysis showed that transitional probabilities were well captured by the data corrected for its bias by the power transformation method (Table 5). Maximum dry and wet period lengths were also projected reasonably well. The average maximum length of a dry period from nine different climate models using conventional bias-correction for Adrian, Fort Wayne, and Norwalk were 24, 27, and 22 days, respectively, when compared to 26, 30, and 25 days in the observed

data (Supplementary Table S7). On the other hand, the maximum wet period length as recorded by the dataset simulated after conventional bias-correction were 19, 21, and 19 days for Adrian, Norwalk, and Fort Wayne, respectively, compared to 9, 11, and 18 days observed (Supplementary Table S7). The possibility of using SWGs for bias-correction cannot be completely discarded because of their ability to better predict the number of wet sequences (Table 5). The number of days for optimum growth of corn and number of snow days were well simulated by the conventional method of bias-correction

(Supplementary Table S7). Air temperature simulations were not improved with any method of bias-correction and descriptive statistical analysis showed that the climate data for maximum and minimum air temperature from the two different sources can be directly use for any changing climate applications without any bias-correction.

The mean values for maximum one-day precipitation during each year showed that conventional bias-correction treatment was able to maintain the mean, skewness, and standard deviation. The range of mean values for one-day maximum

precipitation simulated by each model after treatment with the power transformation method for Adrian, Fort Wayne, and Norwalk, were 57.3-61.2 mm, 57.5-62.2 mm, and 61.6-66.1 mm, when compared to observed values of 55.3 mm, 55.6 mm, and 64.2 mm, respectively (Supplementary Table S4). Even the distributions seen using Lorenz curves (Fig 9) showed that the power transformation reduced the bias and projected similar distributions as those observed from the ground-based stations. With use of the SWGs, Brier score increased. It was 0.6 - 0.7 with CLIGEN and LARS-WG and 0.5 with the

conventional method, which indicated that the SWG simulation outputs were less close to the observed data. Percent correct was also lower when SWGs were used (0.3-0.4) over conventional method (0.5). The bias was higher when using the SWGs (2.6-3.0) compared to the conventional method (1.2-1.4) (Supplementary Table S8).

These results indicated that the power transformation outperformed any other method of bias-correction in this study. Moreover, it was only precipitation which required bias-correction. SWGs have a huge potential in bias-correction. Testing

with parameters other than defaults and understanding the impact of realizations in correcting the bias needs further research but was beyond the scope of this study.

### 3.3 Analysis of climate projections for Western Lake Erie Basin

The Q-Q plots in Figures 6, 7, and 8 showed that daily precipitation depth and average maximum and minimum air temperature could increase. Variance scaling of the air temperature demonstrated large changes at either extreme ends of the

Q-Q plots (Figures 7 and 8), which seems unrealistic and suggested that downscaled data from different climate models can be used without treating it with different bias-correction methods for further studies based on temperature data. In both cases, before and after bias-correction, the future climatic predictions showed that mean precipitation over the WLEB will increase



under both the high (RCP 8.5) and medium (RCP 4.5) emission scenarios for 2006 to 2099. The daily mean precipitation values by 2099 would range between 2.6 and 2.8 mm for different climatic projections (observed range 2.4 - 2.6 mm). The Q-Q plots of corrected daily precipitation for the future time series revealed that there would be more precipitation events and the magnitude of the maximum daily precipitation event would be much greater than the observed data (Figure 6). The

maximum value of daily precipitation after bias-correction was 213.2 mm for RCP 4.5 for Norwalk, OH, compared with 111.8 mm during the baseline period. RCP future climate projections after bias-correction had 53.1-53.5% no precipitation days for Adrian, MI, indicating that the number of days with no precipitation will eventually decrease in the future. The deviation from the mean is anticipated to increase in the future for all three stations in WLEB. For Fort Wayne, the standard deviation for the corrected time series under RCP 4.5 and 8.5, will be 7.7 and 7.9 mm, respectively, compared to 6.7 mm in

the baseline observed data.

The different climate projections after bias-correction showed that maximum daily precipitation would increase for the three stations, and this will be especially evident during summer and spring. The descriptive statistical analysis on a seasonal basis revealed that the number of days with no precipitation will decrease and the mean of average daily precipitation depth will be increasing. For winter, the mean daily precipitation depth was projected to be between 1.9-2.3 mm (1.7 -1.9 mm for

observed data); 2.8-3.2 mm for summer (2.8-3.1 for observed data); 3.2-3.5 mm for spring (3.0-3.2 mm for observed data); and 2.4-2.5 mm for fall (2.3 mm for observed data) for the three stations in this study (Supplementary Table S13).

For fall, maximum daily precipitation depths exceeded the recorded observed values.  The increase in the number of precipitation days will be more pronounced in summers. Number of days with no precipitation decreased to 52.7% for RCP 4.5 and 53.6% for RCP 8.5 compared to 70.7% in the observed baseline data at Adrian, MI for summer, and similar changes

were found for Fort Wayne and Norwalk. The maximum daily precipitation depth received during winters at Fort Wayne will be nearly constant, though there will be more precipitation days. On the other hand, the daily precipitation maximums at Adrian and Norwalk will be greater than that seen in the observed data. This implies that by the end of 21$^{st}$ century precipitation events with higher magnitudes will be more evident. Maximum dry and wet period lengths were projected to increase at all three stations, as well as the number of dry and wet sequences. The number of days for optimum corn growth

will be greater than current levels for WLEB. The snow days were projected to be greater as well for the bias-corrected data but lower for the uncorrected weather data. The mean length of dry periods will be reduced to 3 days from 3-4 days for all stations. The transitional probabilities will be more or less the same, when compared with the observed data, indicating the reliability of the future climatic projections. Table 6 reports different climate indices for Fort Wayne, IN (Supplementary Table S7 for all other stations).

The number of wet days per month is projected to considerably increase and could range between 12 and 16 instead of 8 to 14 days as observed in data from ground-based station in Adrian, MI; 12-15 days (9-13 days in observed baseline data) for




Fort Wayne, IN; and 13-16 days (9-13 days in observed baseline data) for Norwalk, OH. Most changes were seen during

April to October for the number of wet days (Supplementary Table S5 and S6).

Figure 9. Performance evaluation of different bias-correction methods for historic period (1966-2005) in reducing the bias in

5    the daily time series in simulating daily precipitation, mm for Adrian, MI; Fort Wayne, IN; and Norwalk, OH using Lorenz

Curve.





The growing degree days will be sufficient for seeding, flowering and harvesting of corn, but the climate analysis indicated that the growth period may shift 15 days earlier. GDD values for May 1 and 15 varied from 118-144 and 198-236 heat units, respectively, in comparison to 60 and 104 heat units in the observed baseline temperature data for Adrian, MI. GDD values were projected to be 152-178 and 249-287 on May 1 and 15, respectively, compared with 86 and 148 from the observed baseline data for Fort Wayne, IN (Table 6). GDD values at Norwalk, OH were projected to be 124-153 and 209-249 for May 1 and 15, respectively, compared to the observed baseline data values of 80 and 129 (Supplementary Table S9).

Under medium and high emission scenarios, except for Bowling Green and Sandusky, OH, the average maximum dry period length computed from nine different climate projections was expected to increase compared to observed data from the ground-based climate stations, whereas median values from the different climate projections could decrease. For example for Bucyrus, OH, it was estimated that the average and median values of the maximum dry period length were 30 and 23 days from 2066-2099 (RCP 4.5) compared to 25 days observed during the historic period from 1966-2005; average and median values are expected to increase further to 31 and 28 days respectively under RCP 8.5. The maximum wet period length for both the medium or high emission scenarios increased for all eight stations in the WLEB. Maximum wet period length was projected to be between 14 and 30 days long during the 21$^{st}$ century compared to 9 to 18 days recorded from ground-based climate stations for the historic period. The number of dry sequences was projected to increase except in Bowling Green and Sandusky during the 21st century. The range of days for optimal growth of corn may decrease and so may the snow days under the two emission scenarios. However, the number of wet sequences was projected to increase, indicating more frequent precipitation and less dry days through the end of the century. Moreover, with such conditions, it is expected that most precipitation will fall as rain, not snow.

Median values for one-day maximum precipitation were projected to be higher except for Bucyrus and Defiance, OH, under RCP 4.5 where the average and median values for one-day maximum precipitation decreased. The mean value of daily precipitation depth was projected to increase 0.1-0.4 mm for all the stations in the WLEB, and the skewness and standard deviation were also projected to increase. Additionally, higher daily temperatures are anticipated under RCP climate projections for the 8 stations in the WLEB. The rise in average temperature in the WLEB may vary between 1 and 5°C, with more expected under the high emission scenario. One-day value of maximum temperature can increase from a few degrees to +10°C when compared to current climatic conditions. An increase between 1 and 3°C can be seen for maximum values for minimum temperature, but it is topographic specific. Lima, Defiance, and Sandusky all have lower projected one-day maximum values of minimum temperature than current baseline values (Supplementary Table S14, S15, and S16).

**4 Conclusions**

Water is an essential component for human survival and ecosystem sustenance. Movement of water under different future climate projections should be determined as accurately as possible at the regional, national and global scales, to help





determine policies for a sustainable future with sufficient supplies of good quality water (Kalcic et al., 2016;Liu et al., 2016). This can only be achieved if the climate projections for air temperature and precipitation are free from as much bias as possible (Christensen et al., 2007;Teutschbein et al., 2011). Biases in climate projections occur mainly because of flawed or faulty ideational boundary assumptions and can lead to deleterious outcomes. The use of uncorrected climate projections from downscaled climate models in hydrologic modeling or any other applications can lead to lot of uncertainty (Déqué et al., 2007;Kjellström et al., 2011;Mehan et al., 2017b). Therefore, it is always suggested to have a reliable climate database free from most of errors.

Many bias-correction methods exist, including delta change (Hay et al., 2000), linear scaling (Leander and Buishand, 2007;Leander et al., 2008), distribution mapping (Boé et al., 2007;Sennikovs and Bethers, 2009) and other highly efficient methods. There are one or more drawbacks associated with each of the method. Some of them are difficult to understand and implement, require excessive time and computational resource, and/or preserve only the mean. Therefore, out of all conventional methods this study incorporated the use of power transformation and variance scaling to conserve the mean and variance of the weather parameters. Additionally, there was an effort to evaluate the application of SWGs, including CLIGEN and LARS-WG for bias-correction, which was different from conventional statistical downscaling. The outputs from both methods (SWGs and Conventional) were compared with the observed data and used to create the reliable future climatic database for the WLEB.

The outcomes from the study can be summarized as below:

    −    Outputs from the MACA source were better than the GDO source, even without any treatment to correct the bias. Though both datasets were tested and corrected for bias, additional bias was present in the precipitation values that needed to be corrected. The metadata files from the GDO outputs suggested that the historic period was 1950-2015, but it was 1950-2005. Very limited information was available in the source documentation, and care must be taken by users to properly understand the historic period before applying future projections to any modelling application.

    −    Bias-correction using conventional methods, including power transformation and variance scaling, and SWGs (Stochastic Weather Generators), were tested for their effectiveness using distributions, descriptive statistics, and climate indices or extremes. The idea comes from Mehan et al. (2017a), where it was seen that weather generators were successful in capturing descriptive statistics and extremes, while simulating long-term climate at a location provided they are run for an optimum number of realizations to capture the variability in the climate data. Moreover, SWGs help in redistribution of precipitation which is a key element in weather simulation, especially correcting bias during simulations where the large precipitation events are not well captured and the probability of having a wet day increases.





- Conventional method of bias-correction, including power transformation for precipitation outperformed other approaches in this study. Temperatures were less sensitive than precipitation values, which needed further bias-correction.

- SWGs have the potential for bias-correction because of their ability to preserve descriptive statistics and some climate indices. In this study, it was seen that 75th percentile for LARS-WG maintained the mean of daily precipitation, while standard deviation was preserved by the 90th percentile of LARS-WG. The main reason for poor performance of the SWGs in this study was their inability to compute accurate transitional or conditional probabilities. If this can be improved, the overall efficacy of SWGs in bias-correction could also be improved.

- The GCMs underestimated the number of dry days as they were redistributing the amount of precipitation. Alternately, the number of wet days simulated by GCMs in a month was more than what was observed in the station data.

- The future climate projections indicated that the WLEB will have more frequent rainfall events, and annual precipitation totals may increase to nearly double the current levels. Air temperature increase by 1-5°C is projected for the WLEB by the end of 21st century. The crop growing period will shift earlier because of earlier accumulation of sufficient heat units for planting crops.

- This study suggests that the means should not be considered the only criteria to evaluate the performance of any weather simulation forecast or bias-correction method. Other essential characteristics like skewness, standard deviation, and climate extremes should also be evaluated. Verification forecasting and skill scores have huge potential to assess climate projections before the climate values should be put to any further use.

- The results above indicated that precipitation was the key element of all three weather variables which undergoes the most changes during the climate simulation process. Temperature simulations were affected least during downscaling and bias-correction. The prime reason for the establishment of climate models was to simulate the dynamics of aerosols and the amount of greenhouse gases that contribute to global warming (Moriondo et al., 2016;Xu, 1999). Therefore, the precision of the climate models in simulating the air temperature was better than precipitation. Henceforth, precipitation data should be thoroughly analyzed for its bias and should be corrected before it can be used for any hydrologic and/or crop modeling studies, as precipitation is a critical factor as it forms an important component of the hydrologic cycle.

- The results from this study were very useful in creating a reliable climate database for the entire WLEB, which can be used in further hydrologic assessment studies looking at the impact of changing climatic patterns on water quality in Lake Erie.





**Supplemental Data.** Provided as a separate file uploaded with this manuscript.

**Data availability.** Data used in this study were obtained from two sources: 1.) GDO (authors created acronym for Global Downscaled Climate and Hydrology Projections), available at the URL: https://gdo-dcp.ucllnl.org/downscaled_cmip_projections/; and 2.) MACA (Multivariate Adaptive Constructed Analogs), available at the URL: https://climate.northwestknowledge.net/MACA/. Daily summaries of climate data from ground based climate stations were downloaded from https://www.ncdc.noaa.gov/cdo-web/datasets.

**Author Contribution.**

Sushant Mehan designed the study and performed analysis and wrote the initial drafts of the manuscript under the guidance of Dr. Margaret W. Gitau who supervised the project. Dr. Dennis C Flanagan, who is development leader for CLImate GENerator (CLIGEN), a stochastic weather generator used in this study, provided insights on the results and contributed to the final manuscript.

**Competing interests.** There is no competing interest.

**Acknowledgements.**

This work is supported in part by the USDA National Institute of Food and Agriculture, Hatch Project 1009404. Authors want to thank you Qi Wang from Statistical Software Consulting, Purdue University for providing some insights with R language.

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





## Tables

Table 1. Different GCM models used for quantifying the error or bias when compared with the ground-based station from NOAA's Climate Data Online facility.

| S. No. | GCM | Basic Source | Studies based on source |
|---|---|---|---|
| 1 | Beijing Climate Center Climate System Model, Beijing, China (**BCCCSM**) | http://forecast.bcccsm.ncc-cma.net/htm/ | (Friedlingstein et al., 2014;Sun et al., 2015) |
| 2 | Community Climate System Model, USA (**CCSM4**) | http://www.cesm.ucar.edu/ | (Lawrence et al., 2012;Palazzoli et al., 2015) |
| 3,4 | Geophysical Fluid Dynamic Laboratory, USA (**GFDL_ESM2G and GFDL_ESM2M**) | http://nomads.gfdl.noaa.gov:8080/DataPortal/cmip5.jsp | (Straatsma et al.;Gorguner et al.;Sridhar et al., 2017) |
| 5,6 | Institute Pierre Simon Laplace Climate Modeling Center, France (**IPSL_CM5ALR and IPSL_CM5AMR**) | http://icmc.ipsl.fr/ | (Murawski et al., 2016;Song et al., 2017) |
| 7,8 | **MIROCESM** and **MIROCESMCHEM**, Japan | http://www.geosci-model-dev.net/4/845/2011/ | (Peng et al., 2016;Park et al., 2014) |
| 9 | Norwegian Earth System Model, Norway (**NorESM1M**) | http://adsabs.harvard.edu/abs/2013GMD.....6..687B | (Lant et al., 2016;Ivancic, 2016) |

5  Table 2 List with explanation, application, and computational formula for various climate indices, verification skill scores, and performance coefficients.

| CLIMATE INDICES | | |
|---|---|---|
| **Parameter Name** | **Definition** | **Application** |
| Count of Dry Spell (Mathugama and Peiris, 2011;Sivakumar, 1992;Taley and Dalvi, 1991;Mathlouthi and Lebdi, | A period with at least 15 consecutive days in which none of the days had greater than 0.1 mm of rainfall | Onset and cessation of droughts can be projected using the count of dry spells. Moreover, dry spells affect aquatic biodiversity, crop growth, and hydropower generation. |





---

2008;Douguedroit, 1987)

| | | |
|---|---|---|
| Count of Wet Spell (Bai et al., 2007) | A period with more than three wet days (precipitation more than or equal to 0.1 mm) ending with two continuous dry days (precipitation less than 0.1 mm) | Information on wet spells is important for optimizing water allocation and distribution, which is instrumental in planning flood control remedies and regulating sediment yield into the main streams. |
| Number of dry and wet day count in a month | Absolute number of days with precipitation depth of less than and at least 0.1 mm, respectively on a single day | Estimates of crop water requirements and decisions on when to plant depend on monthly numbers of dry and wet days. |
| Number of Snow Days (Auer, 1974;Lawrence, 2015) | Day precipitation depth more than 0.1 mm and average temperature lower than 2°C. | The water budget of snow-dominated watersheds is dependent on the count of snow days. |
| Growing Season Requirement/ Period of optimal growth (Neild and Newman, 1987;Loftus, 1999;Davis, 2012) | Number of days during the entire crop growing season with average temperatures between 20 and 25 °C (supports corn growth in Midwest USA) | Estimation of growth and yield of corn requires information on period of optimal growth of corn |
| Growing Degree Days (GDD) or Heating Units (HU) (Neild and Newman, 1987) | Heating Units (HU) indicate energy accumulations affecting crop cycles from planting through to harvesting. | $$GDD = \frac{Maximum\,Temperature + Minimum\,Temeprature}{2} - Base\,Temperature$$ Different stages of the crop growth cycle can be simulated using information on Heating Units (HU). |
| Count for Maximum Dry and Wet Length (Deni et al., 2008) | The longest continuous stretch of the dry and wet period | Information on this data helps in identifying extreme events, including dry and wet period |
| Probability of dry day (Pd) Probability of wet day (Pw) Probability of dry followed by dry day(P (D\|D)) | All these factors are critical in generating long-term climate simulation, hence needed evaluation. Moreover, mean length of dry and wet period decides onset of planting and harvesting in rainfed | $$P_d = \frac{Number\,of\,dry\,days}{Total\,number\,of\,days}$$ $$P_w = \frac{Number\,of\,wet\,days}{Total\,number\,of\,days} = 1 - P_d$$ $$P_{d|d} = \frac{Number\,of\,sequence\,with\,two\,dry\,days}{Total\,number\,of\,dry\,days}$$ |





| Probability of wet day followed by wet day (P (W\|W)) | agricultural places. (for the transition probabilities computation, the dry day is day with the precipitation 0.01 mm and anything equal and more than 0.01 mm is wet day for all other purposes, the threshold 0.1 mm) | $P_{w\|w} = \dfrac{\text{Number of sequence with two wet days}}{\text{Total number of wet days}}$ |
|---|---|---|
| Probability of wet day followed by dry day (P (W\|D)) | | $P_{w\|d} = 1 - P_{d\|d}$ |
| Probability of dry day followed by wet day (P (D\|W)) | | $P_{d\|w} = 1 - P_{w\|w}$ |
| Average length of dry and wet day (Ld, and Lw) | | $L_d = \dfrac{1}{P_{w\|d}}$ <br> $L_w = \dfrac{1}{1 - P_{w\|w}}$ |
| Return time Period to have an event equal to average length of dry and wet day (Td and Tw) (Pandharinath, 1991;Sonnadara and Jayewardene, 2015) | | $T_d = \dfrac{1 - P_{w\|w} + P_{w\|d}}{\text{Number of days in a month} * P_{w\|d}(1 - P_{w\|w})(1 - P_{w\|d})^{L_d}}$ <br><br> $T_w = \dfrac{1 - P_{w\|w} + P_{w\|d}}{\text{Number of days in a month} * P_{w\|d}(1 - P_{w\|w})P_{w\|w}{}^{L_d}}$ |
| One day maximum Precipitation (mm) (Bhattacharaya and Sarkar, 1982;Upadhaya and Singh, 1998) | Maximum value of single day precipitation event | Drainage design, soil conservation and management, risk mitigation, in events, including flash floods and droughts |

**VERIFICATION PARAMETERS**

| Parameter Name | Definition | Formula | Range |
|---|---|---|---|
| Lorenz Curve (Gastwirth, 1971) | Daily precipitation totaled data is arranged in increasing order, cumulative, and converted to a proportion of total precipitation | | |
| Gini Coefficient (advantage over the other measures of variability such as | Two times the area between line representing uniform precipitation distribution (slope = 1) and Lorenz curve | $G = \dfrac{1}{n}\left(n + 1 - 2\left(\dfrac{\sum_{i=1}^{n}(n+1-i)y_i}{\sum_{i=1}^{n}y_i}\right)\right)$ | [0,1] <br> 0 represent a uniform distribution |



| | | | |
|---|---|---|---|
| standard deviation, which are scale and probability dependent and make them less robust) (Gastwirth, 1972) | | Where yi indicates PRCPTOT or SD II at a particular year i and n indicated total number of years. PRCPTOT is defined as the total amount of precipitation on wet days (days with precipitation >0.1 mm). SDII is the annual precipitation intensity, obtained by dividing the total amount of precipitation annually by the count of days in a year with precipitation depth more than 0.1 mm | over the time period and 1 represents all the precipitation occurred on a single day |
| LEPS (Linear Error in Probability Space) Score (Potts et al., 1996) | LEPS is independently sensitive to bias and forecast variance if forecast is less than observed. LEPS are less sensitive to outliers than correlations but more sensitive to changes in values near the center of the cumulative probability distribution. It can be used to assess the forecast of continuous and categorical variables. LEPS score is equitable and does not "bend back" (give better scores for worse forecasts near the extremes) | $$LEPS = 3(1 - |CDF(sim) - CDF(Obs)| \\ + CDF(sim)^2 \\ - CDF((sim) + CDF(obs)^2) - 1$$ <br><br> Where CDF is cumulative probability density function | Range: 0 to 1. Perfect score: 0 |
| Brier Score (Murphy, 1973) | Measures the mean squared probability error | $$BS = \frac{1}{n} \sum_{i=1}^{n} (f_i - o_i)^2$$ <br><br> Where fi are forecast probabilities between 0 and 1 and oi are given as 0 and 1 for observed dry and wet day respectively. | Lower brier score means the forecast is closer to the observation. BS can be partitioned into three terms: (1) reliability, (2) resolution, and (3) uncertainty. |
| Heidke Skill Score (HSS) (Heidke, 1926) | Ratio of difference between the number of times the forecast matches with the observation and number of categorical correct forecast and difference | $$Heidke = \frac{(H - E)}{(N - E)}$$ <br><br> Where H is the number of categorical forecasts (hits), N is total number of forecast issue. E is the number of | Perfect score is 1 (perfect set of forecasts) Random forecast would |



| | | categorically correct forecast | be scored as 0. Set of forecast having fewer hits would be negative scores. |
|---|---|---|---|
| Pierce Skill Score (PSS) (Peirce, 1884;Hanssen and Kuipers, 1965;Murphy and Katz, 1985;Flueck) | PSS measure the difference between probability of detection and false detection. In other words, it measures the ability of the forecast to differentiate between occurrence of an event or not. | $$PSS = H - F$$ Where, H = Hit Rate (Relative number of times an event was forecast when it occurred) and F is False Alarm Rate (relative number of times the event was forecast when it did not occur) | Range: -1 to 1, 0 indicates no skill. Perfect score: 1 |
| Percentage Correct (Finley, 1884) | Forecast accuracy by considering the simple matching coefficient based on the ''proportion'' of total ''correct'' hits and rejections (PC) | $$PC = \frac{B - H - FB + 2HF}{B + F - H}$$ Where B = Bias, H = Hit rate, F = False Alarm Rate | |
| Bias (Finley, 1884) | Verification metric denoted by ratio of total number of events forecast and total number of events observed, Forecast is termed as under forecast when BIAS<1 or over forecast (BIAS>1) events. | $$Bias = \frac{h + f}{h + m}$$ Where h = Hit, f = False Alarm, m = miss | Perfect Score: 1 |
| Odd's Ratio Skill Score Yule's Q (Stephenson, 2000;Yule, 1900) | Because ORSS is independent of the marginal distribution, it strongly discriminates between the cases with and without association even when the different contingency tables appear to have similar cell counts. So is difficult to hedge. | $$ORSS = \frac{H - F}{H + F - 2HF}$$ Where, H = Hit Rate F = False Alarm Rate | [-1,1] 0 indicates no skill Perfect score 1 |
| Extreme Dependent Score (Ferro and Stephenson, 2011) | EDS is independent of bias, so should be presented together with the frequency bias. | $$EDS = \frac{\ln p - \ln H}{\ln p + \ln H}$$ Where p=(hits+misses)/total is the base rate (climatology), q=(hits+false alarms)/total is the frequency with which the event is forecast, H is the hit rate, also known as the probability of detection, and F is the false | [-1, 1], 0 indicating no skill with 1 representing perfect score. |



alarm rate, also known as the probability of false detection.

| Performance Coefficient | | | |
|---|---|---|---|
| **Parameter Name** | **Definition** | **Formula** | **Range** |
| NSE (Nash and Sutcliffe, 1970) | It is usually the coefficient to assess the predictive power of the simulation models, most frequently used in hydrology | $NSE = 1 - \dfrac{\sum_{i=1}^{n}\left(X_{obs}^{i} - X_{sim}^{i}\right)^2}{\sum_{i=1}^{n}\left(X_{obs}^{i} - \overline{X_{obs}}\right)^2}$ | $[-\infty, 1]$, with 1 as perfect score; 0 means the projections are as accurate as the mean of the observed data, anything less than 0 means accuracy of simulation is being compromised. |
| Coefficient of Correlation (Galton, 1889) | Linear measure of observed and simulated values. It does not take forecast bias into account and is sensitive to outliers | $RC = \dfrac{\sum_{i=1}^{n}\left(X_{obs}^{i} - \overline{X_{obs}}\right)\left(X_{sim}^{i} - \overline{X_{sim}}\right)}{\sqrt{\sum_{i=1}^{n}\left(X_{obs}^{i} - \overline{X_{obs}}\right)^2 \sum_{i=1}^{n}\left(X_{sim}^{i} - \overline{X_{sim}}\right)^2}}$ | $[-1, 1]$ with 1 being the perfect score |
| Relative Error | Relative error is function of absolute error and the observed value and expressed as absolute error divided by the magnitude of the exact value. | $RE = \dfrac{\sum_{i=1}^{n} X_{sim}^{i} - \sum_{i=1}^{n} X_{obs}^{i}}{\sum_{i=1}^{n} X_{obs}^{i}} \times 100 \ (\%)$ | $[-\infty, \infty]$, values near to zero is considered to be better. |
| Cohen's-d effective size (Cohen, 1988;Glass et al., 1981;Cohen, 1977) | Alternate measure of checking the difference in mean distributions | $Cohen's\ d = \dfrac{M1 - M2}{SD_{cont}}$  <br><br> Where M1 and M2 are means from the simulated and observed data and SD control is standard deviation from observed data. | $[0\text{-}1]$, where values closer to 0 are considered better for good simulation. |





Table 3 Comparison of GDO and MACA climate projection sources for Adrian, MI, Fort Wayne, IN, and Norwalk, OH in simulating descriptive statistics for daily precipitation (mm), and maximum and minimum air temperatures (°C)

| Precipitation, mm | | | | | | | | |
|---|---|---|---|---|---|---|---|---|
| | Adrian, MI | | | Fort Wayne, IN | | | Norwalk, OH | | |
| Treatment | Median | NDP0* (%) | Maximum | Median | NDP0 (%) | Maximum | Median | NDP0 (%) | Maximum |
| Observed | 0 | 66.9 | 120.4 | 0 | 63.5 | 111.8 | 0 | 64 | 229.1 |
| GDO No Treatment | (0.2-0.2),0.2 | (29.8-31.9),30.9 | (65.4-110.1),83.3 | (0.4-0.5), 0.4 | (15.4-20.7), 17.7 | (52.0-72.0), 63.7 | (0.7-0.8), 0.8 | (10.8-13.0), 12.0 | (40.1-48.0), 43.6 |
| MACA No Treatment | (0.0-0.0),0.0 | (53.5-54.1),53.9 | (67.2-71.0),69.7 | (0.0-0.0),0.0 | (54.6-55.5), 54.9 | (65.0-74.5), 72.3 | (0.0-0.0)0.0 | (51.0-51.7), 51.4 | (54.5-112.8), 101.6 |
| Maximum Temperature, °C | | | | | | | | |
| Treatment | NDT35 † (%) | Maximum | Minimum | NDT35 (%) | Maximum | Minimum | NDT35 (%) | Maximum | Minimum |
| Observed | 0.3 | 40.0 | -20.0 | 0.3 | 41.1 | -23.9 | 0.2 | 39.4 | -22.2 |
| GDO No Treatment | (0-0.4), 0.2 | ( 36.0-38.9 ), 37.4 | ( -20.4--16.2 ), -18.5 | (0.1-0.5), 0.3 | (36.7-39.5 ), 38.1 | ( -23.2--17.4 ), -19.7 | (0-0.3), 0.1 | (34.8-39.8), 36.9 | ( -21.3--14.9 ), -18.2 |
| MACA No Treatment | (0.5-0.7), 0.6 | ( 39.5-40.2 ), 39.9 | ( -17.5--16.5 ), -17.1 | (0.5-0.8), 0.7 | (40.6-42.1 ), 41.8 | ( -22.1--20.3 ), -21.4 | (0.2-0.3), 0.2 | ( 37.6-37.8 ), 37.7 | ( -19.2--17.8 ), -18.8 |
| Minimum Temperature, °C | | | | | | | | |
| Treatment | NDT2‡ (%) | Maximum | Minimum | NDT2 (%) | Maximum | Minimum | NDT2 (%) | Maximum | Minimum |
| Observed (MACA) | 46.3 | 24.4 | -30.0 | 41.1 | 25.6 | -30.0 | 41.8 | 25.0 | -29.4 |
| Observed (GDO) | 45.9 | 24.4 | -30.0 | 40.8 | 25.6 | -30.0 | 41.6 | 26.1 | -29.4 |
| GDO No Treatment | (44.4-46.0), 45.4 | ( 21.7-26.3 ), 23.6 | ( -31.2-25.8 ), -29.0 | (39.5-41.2), 40.3 | (22.7-26.8), 25.0 | ( -33.8-26.8 ), -30.2 | (39.9-41.6), 40.9 | ( 22.0-27.8), 24.6 | ( -29.7--23.6 ), -27.2 |
| MACA No Treatment | (44.8-45.7), 45.3 | ( 23.8-24 ), 24 | ( -28.2--26.4 ), -27.9 | (39.9-40.7), 40.4 | ( 25.2-25.5 ), 25.5 | ( -28.9--26.9 ), -28.4 | (41-41.6), 41.4 | ( 24-24 ), 24 | ( -28--27 ), -27.5 |

**\*** Number of days with no precipitation expressed as percentage of the total dataset NDP0 (%)

† Number of days with maximum temperature more than 35°C expressed as percentage of the total dataset NDT35 (%)

‡ Number of days with maximum temperature more than 2°C expressed as percentage of the total dataset NDT2 (%)





Table 4 Comparison of GDO and MACA climate projection sources for different climate indices for Adrian, MI, Fort Wayne, IN, and Norwalk, OH for period between 1966 and 2005

| Adrian, MI | | | | | | |
|---|---|---|---|---|---|---|
| **Treatment** | P(W\|W) | P(W\|D) | Ld | Lw | Td | Tw |
| **Observed** | 0.5 | 0.3 | 4 | 1 | 1 | 4 |
| **GDO No Treatment** | (0.7-0.7), 0.7 | (0.5-0.6), 0.5 | (2-2), 2 | (2-2), 2 | (13-33), 21 | (1-1), 1 |
| **MACA No Treatment** | (0.6-0.6), 0.6 | (0.3-0.4), 0.3 | (3-3), 3 | (2-2), 2 | (2-3), 2 | (1-2), 1 |
| **Fort Wayne, IN** | | | | | | |
| **Observed** | 0.5 | 0.3 | 3 | 1 | 1 | 3 |
| **GDO No Treatment** | (0.8-0.8), 0.8 | (0.6-0.7), 0.6 | (2-2), 2 | (3-3), 3 | (52-174), 81 | (1-1), 1 |
| **MACA No Treatment** | (0.6-0.6), 0.6 | (0.3-0.3), 0.3 | (3-3), 3 | (1-2), 2 | (2-2), 2 | (1-2), 1 |
| **Norwalk, OH** | | | | | | |
| **Observed** | 0.5 | 0.3 | 3 | 1 | 1 | 3 |
| **GDO No Treatment** | (0.9-0.9), 0.9 | (0.7-0.8), 0.7 | (1-2), 1 | (3-4), 4 | (256-2016), 738 | (1-1), 1 |
| **MACA No Treatment** | (0.6-0.6), 0.6 | (0.3-0.4), 0.4 | (3-3), 3 | (2-2), 2 | (2-3), 3 | (1-1), 1 |

5    Table 5. Extreme event analysis/climate indices analysis for Adrian, MI from the MACA climate projection before and after different bias-correction methods compared with observed climate data from ground-based station

| | Maximum Dry Period Length (days) | Maximum Wet Period Length (days) | Number of dry sequences | No. of wet sequences | No. of days for optimum growth of corn | Snow Days |
|---|---|---|---|---|---|---|
| **Observed** | 26 | 9 | 33 | 153 | 51 | 30 |
| **MACA No Treatment** | (17-29), 22 | (16-23), 19 | (4-17), 11 | (318-450), 387 | (57-61), 60 | (40-43), 42 |
| **MACA Conventional** | (17-32), 24 | (15-23), 19 | (4-19), 12 | (314-446), 381 | (28-30), 29 | (60-62), 61 |
| **MACA CLIGEN 75** | (13-36), 23 | (175-228), 210 | (0-5), 2 | (108-170), 142 | (0-0), 0 | (332-338), 335 |
| **MACA CLIGEN 90** | (1-3), 2 | (2146-3296), 2845 | (0-0), 0 | (0-1), 0 | (0-0), 0 | (365-365), 365 |
| **MACA LARS-WG 75** | (3-4), 3 | (209-572), 297 | (0-0), 0 | (20-58), 34 | (73-80), 77 | (98-106), 102 |
| **MACA LARS-WG 90** | (1-1), 1 | (2460-7901), 5183 | (0-0), 0 | (0-0), 0 | (73-80), 77 | (104-110), 108 |