# Peer review of "Development of reliable future climatic projections to assess hydrometeorological implications in the Western Lake Erie Basin"

_Hydrology and Earth System Sciences, 2018_

## Referee Comment (RC1) · M. Zappa (Referee) · 3 Sep 2018

This manuscript presents a data set of climate forcing thought to be used for hydrological impact studies.

The authors carefully present the data sources and adopt different methodologies to treat them. A large selection of established metrics are introduced and used for evaluating the data with focus on three particular locations within the Western Lake Erie Basin target area. The three locations correspond to sites with ground truth data on precipitation and air temperature (incl. MIN and MAX).

The reading of the manuscript is quite straight forward. It is fore sure a meaningful piece of information and surely a nice initiative by the authors to make their data public. It is not usual to have data papers in HESS, but there are now and then quite interesting examples such as the CAMELS data set by Addor et al. (2017). While it is quite obvious to me how incremental and novel Addor et al. (2017) is, I struggle in finding the same level of novelty in this manuscript. Several countries develops their set of bias corrected climate forcing for applications in hydrology and other sciences including validations with respect to ground truth.
In Switzerland the current data set (http://www.ch2011.ch/en/index.html) will be soon updated with a new generation of scenarios (http://www.ch2018.ch/en/home-2/).

At the beginning of the annotated manuscript I attached, I warmly suggest to consider moving this manuscript to another Copernicus Journal. "Earth System Science Data" (https://www.earth-system-science-data.net/) is to me the right platform to present this piece of work. According to the list of "manuscript types" for HESS (https://www.hydrology-and-earth-system-sciences.net/about/manuscript_types.html), there is the option to have a "cutting-edge case study" accompanying the data. This is not the case for the present manuscript and thus I suggest contemplating this option in a next phase of the review process.

Best regards

Massimiliano Zappa
Birmensdorf, 3.9.2018

Reference:
Addor, N., Newman, A. J., Mizukami, N., and Clark, M. P.: The CAMELS data set: catchment attributes and meteorology for large-sample studies, Hydrol. Earst Syst.

Sci., 21, 5293-5313, https://doi.org/10.5194/hess-21-5293-2017, 2017.

Please also note the supplement to this comment:
https://www.hydrol-earth-syst-sci-discuss.net/hess-2018-204/hess-2018-204-RC1-supplement.pdf

---

## Referee Comment (RC2) · L.A. Melsen (Referee) · 17 Sep 2018

This paper describes the process of developing a climate database for the Western Lake Erie Basin. This process involves the comparison of two sources of climate projections, and the comparison of different bias-correction methods.

Being a hydrological modeller, I am well aware of the large uncertainties involved with climate projections, and studies to investigate, understand, and – hopefully at some point – diminish these uncertainties is urgently needed. Unfortunately, this study did, at least to me, not substantially contribute to understanding these uncertainties.

MAJOR

The overall structure of the manuscript is clear. However, particular sections and paragraphs require restructuring. The introduction contains many sentences that are not placed in context. For example, pollution mitigation strategies are (suddenly and abruptly) introduced, and not referred back to – what was the motivation to discuss this specific topic? Also Western Lake Erie Basin is not introduced, but is – apparently – a hotspot that requires special attention (see p.3, l. 3, '.. specific to the WLEB'). Another example is that 'climate projections at regional scales are unclear' (p.1, l.31), what is meant by "unclear"? The introduction is the foundation of the paper, but right now the motivation and the problem statement are not clear.

Also in the study itself, many choices were not rationalized. For example, why where three out of eight stations used (p. 3, l. 19) and not all, and how does this influence the results? Same concerning the GCMs, I understand that using all might be a lot, but why nine, and why these nine, and how does this influence the results?

This relates to another point; the discussion is currently not well embedded in scientific literature, and therefore does not lead to deeper understanding of the results. For example; only 9 GCMs were used, are they from different 'families' as discussed in the model genealogy of Knutti et al.? and if not, your ensemble is probably too narrow; how would this influence the results and the conclusions of the study? Another example: written on p. 21: 'biases in climate projections occur mainly because of flawed or faulty ideational boundary assumptions and can lead to deleterious outcomes'. The first question from a skeptic could be if these projections have any value at all; can you correct for faulty assumptions simply with using a bias correction or is this just a Band-Aid? or could that be a motivation to opt for SWGs? As uncertainty is one of the topics dealt with in this paper, a more comprehensive discussion of the approaches and assumptions in this study is well in place, or even needed.

MINOR

Some of the methods of the study are presented as conclusions, such as the points at page 22 starting at line 17 and starting at line 20, while these points are actually motivations or methodologies, and not the result or conclusion of this study per se.

Concerning the text; currently the text contains many numbers, which does not necessarily makes it attractive to read (e.g. p. 12, line 14/15). On the other hand, the figures are sometimes not comprehensively discussed (sometimes referred to only once). Consider removing too many individual values from the text, and sketch a more general picture, refer to figures / tables for detailed numbers.

Overall, I recognize that the study has been done carefully, but scientifically, discussion and depth are missing and (maybe because of that) few new lessons are learned.

---

## Author Comment (AC1) · 28 Oct 2018

Comment 1a: The reading of the manuscript is quite straight forward. It is for sure a meaningful piece of information and surely a nice initiative by the authors to make their data public. It is not usual to have data papers in HESS, but there are now and then quite interesting examples such as the CAMELS data set by Addor et al. (2017). While it is quite obvious to me how incremental and novel Addor et al. (2017) is, I struggle in finding the same level of novelty in this manuscript. Several countries develops their set of bias corrected climate forcing for applications in hydrology and other sciences including validations with respect to ground truth. In Switzerland the current data set

(http://www.ch2011.ch/en/index.html) will be soon updated with a new generation of scenarios (http://www.ch2018.ch/en/home-2/).

Response: The manuscript presents the problems in projected meteorological data based on simulated historic climate values. The methodology and framework were developed using three stations but the future climate projections were made for all eight stations shown in Figure 1 of the manuscript. Using the methodology and comparative outcomes based on this study, we build daily data for all the stations in WLEB, which included 16 ground-based climate stations. In evaluating GCM data from two different databases we found some critical issues and, in particular, that precipitation extremes (which are the ones that present the greatest challenges with respect to management and adaptation) were largely misrepresented. Thus, in addition to quantifying the errors, we also corrected for bias. The corrected data are available as values (rather than changes) at a daily time step and can be used reliably in a variety of applications, and in particular with hydrologic and water quality modeling studies for which daily data are needed. These data are currently published with embargo pending publication of this manuscript and will be available to the public through the Purdue University Research Repository (PURR) in due course.

Comment 1b: At the beginning of the annotated manuscript I attached, I warmly suggest to consider moving this manuscript to another Copernicus Journal. "Earth System Science Data" (https://www.earth-system-science-data.net/) is to me the right platform to present this piece of work. According to the list of "manuscript types" for HESS (https://www.hydrology-and-earth-system-sciences.net/about/manuscript_types.html), there is the option to have a "cutting-edge case study" accompanying the data. This is not the case for the present manuscript and thus I suggest contemplating this option in a next phase of the review process.

Response: We would like to thank the reviewer for this suggestion. This is a good suggestion. However, we'd be concerned about being able to reach our target audience who would be more aligned with HESS. The objectives of our manuscript are aligned
with the Hydrometeorology subject area (stochastic processes, modelling approaches, and uncertainty analysis sub-areas) of HESS hence our choice of the journal. We are willing to consider ESSD if the editor(s) feel that that would be a better fit.
* * *

---

## Author Comment (AC2) · 29 Oct 2018

Comment 1a: The overall structure of the manuscript is clear. However, particular sections and paragraphs require restructuring. The introduction contains many sentences that are not placed in context. For example, pollution mitigation strategies are (suddenly and abruptly) introduced, and not referred back to – what was the motivation to discuss this specific topic? Also Western Lake Erie Basin is not introduced, but is – apparently – a hotspot that requires special attention (see p.3, l. 3, '.. specific to the WLEB'). Another example is that 'climate projections at regional scales are unclear' (p.1, l.31), what is meant by "unclear"? The introduction is the foundation of the paper,

but right now the motivation and the problem statement are not clear.

Response: We appreciate this comment and will revise the manuscript accordingly.

Comment 1b: Also in the study itself, many choices were not rationalized. For example, why where three out of eight stations used (p. 3, l. 19) and not all, and how does this influence the results? Same concerning the GCMs, I understand that using all might be a lot, but why nine, and why these nine, and how does this influence the results?

Response: The selection of three out of eight stations was based on analysis of similarities or differences among the stations in terms of climatic patterns and statistical properties of precipitation and temperatures, and also considering their spatial locations so as to provide suitable representation of climate across the basin, as detailed in (Mehan et al., 2017a). We have added the aforementioned text to the manuscript for clarification. The three stations were used to develop the methodology and framework for this study, which was then applied to all 16 ground-based climate stations in the WLEB. Because of the care taken in identifying the three stations, we believe that the selection did not adversely affect the results but, rather, made the process more efficient.

Concerning the GCMs, these were selected because they were common to both climate databases, thus, allowing a comparison between the databases. While we did not evaluate how this choice influenced the result, we believe that we were able to capture range and variability in climate and thus provide suitable representation with these nine GCMs (with corrections).

(Mehan, S., Guo, T., Gitau, M. W., and Flanagan, D. C.: Comparative study of different stochastic weather generators for long-term climate data simulation, Climate, 5, 26, 2017a.)

Comment 1c: This relates to another point; the discussion is currently not well embedded in scientific literature, and therefore does not lead to deeper understanding of

the results. For example; only 9 GCMs were used, are they from different 'families' as discussed in the model genealogy of Knutti et al.? and if not, your ensemble is probably too narrow; how would this influence the results and the conclusions of the study? Another example: written on p. 21: 'biases in climate projections occur mainly because of flawed or faulty ideational boundary assumptions and can lead to deleterious outcomes'. The first question from a skeptic could be if these projections have any value at all; can you correct for faulty assumptions simply with using a bias correction or is this just a Band-Aid? or could that be a motivation to opt for SWGs? As uncertainty is one of the topics dealt with in this paper, a more comprehensive discussion of the approaches and assumptions in this study is well in place, or even needed.

Response: We would like to thank the reviewer for this comment. We will expand our discussion to provide a better understanding of the results.

Comment 2a: Some of the methods of the study are presented as conclusions, such as the points at page 22 starting at line 17 and starting at line 20, while these points are actually motivations or methodologies, and not the result or conclusion of this study per se.

Response: The conclusions section has been revised based on this and other comments.

Comments 2b: Concerning the text; currently the text contains many numbers, which does not necessarily makes it attractive to read (e.g. p. 12, line 14/15). On the other hand, the fig- ures are sometimes not comprehensively discussed (sometimes referred to only once). Consider removing too many individual values from the text, and sketch a more general picture, refer to figures / tables for detailed numbers.

Response: We appreciate this comment and will revise the manuscript accordingly

Comments 2c: Overall, I recognize that the study has been done carefully, but scientifically, discussion and depth are missing and (maybe because of that) few new lessons

are learned.

Response: We would like to thank the reviewer for this comment. We will revise the manuscript to expand our discussion based on this and other comments provided.